# Domain Adaptation for Sentiment Analysis Using Robust Internal Representations

**Mohammad Rostami**
University of Southern California
rostamim@usc.edu

**Digbalay Bose**
University of Southern California
dbose@usc.edu

**Shrikanth Narayanan**
University of Southern California
shri@usc.edu

**Aram Galstyan**
University of Southern California
galstyan@usc.edu

## Abstract

Sentiment analysis is a costly yet necessary task for enterprises to study the opinions of their customers to improve their products and to determine optimal marketing strategies. Due to the existence of a wide range of domains across different products and services, cross-domain sentiment analysis methods have received significant attention. These methods mitigate the domain gap between different applications by training cross-domain generalizable classifiers which relax the need for data annotation for each domain. We develop a domain adaptation method which induces large margins between data representations that belong to different classes in an embedding space. This embedding space is trained to be domain-agnostic by matching the data distributions across the domains. Large interclass margins in the source domain help to reduce the effect of "domain shift" in the target domain. Theoretical and empirical analysis are provided to demonstrate that the proposed method is effective.

## 1  Introduction

The main goal in sentiment classification is to predict the polarity of users automatically after collecting their feedback using online shopping platforms, e.g., Amazon customer reviews. A major challenge for automatic sentiment analysis is that polarity is expressed using completely dissimilar terms and phrases in different domains. For example, while terms such as "fascinating" and "boring" are used to describe books, terms such as "tasty" and "stale" are used to describe food products. As a result of this discrepancy, an NLP model that is trained for a particular domain may not generalize well in other different domains, referred as the problem of "domain gap" (Wei et al., 2018). Since generating annotated training data for all domains is expensive and time-consuming (Rostami et al., 2018), cross-domain sentiment analysis has gained significant attention recently (Guo et al., 2020; Du et al.,

2020; Gong et al., 2020; Xi et al., 2020; Dai et al., 2020; Lin et al., 2020; Long et al., 2022; Badr et al., 2022; Ryu et al., 2022; Huang et al., 2023). The goal is to relax the need for annotation via transferring knowledge transfer from another domain with annotated data to train models for domains with unannotated data.

The above problem has been studied more broadly in the *domain adaptation* (DA) literature (Wang and Deng, 2018). A common DA approach is to align the distributions of both domains in a shared embedding space (Redko and Sebban, 2017). As a result, a source-trained classifier that receives its input from the embedding space will generalize well in the target domain. In the sentiment analysis problem, this means that polarity of natural language can be expressed independent of the domain in the embedding space to transcendent discrepancies. We can model the embedding space using a shared deep encoder trained to align the distributions at its output space. This training procedure has been implemented indirectly using adversarial learning (Li et al., 2019; Dai et al., 2020; El Mekki et al., 2021; Jian and Rostami, 2023) or by directly minimizing loss functions that are designed to align two probability distributions (Kang et al., 2019; Guo et al., 2020; Xi et al., 2020; Lin et al., 2020; Rostami and Galstyan, 2023a).

**Contributions:** Our idea is based on learning a parametric distribution for the source domain in a cross-domain embedding space (Rostami and Galstyan, 2023b). We estimate this distribution as a Gaussian mixture model (GMM). We then use the GMM distribution to align the source and the target distributions using confident samples, drawn for them GMM to increase the interclass margins to reduce the domain gap.

## 2  Related Work

While domain adaptation methods for visual domains usually use generative adversarial networks

(GANs) (Goodfellow et al., 2014),the dominant approach for cross-domain sentiment analysis is to design appropriate loss functions that directly impose domain alignment. The main reason is that natural language is expressed in terms of discrete values such as words, phrases, and sentences. Since this domain is not continuous, even if we convert natural language into real-valued vectors, it will not be differentiable. Hence, adversarial learning procedure cannot be easily adopted for natural language processing (NLP) applications. The alternative approach is to minimize a probability distribution metric to reduce domain cap (Shen et al., 2018). For example, using Wasserstein distance (WD) for domain alignment in visual domains has been found to be highly effective (Long et al., 2015; Sun and Saenko, 2016; Lee et al., 2019; Rostami et al., 2019). We rely on the sliced Wasserstein distance (SWD) for aligning distribution because of having a less computational load (Lee et al., 2019).

The major reason for performance degradation of a source-trained model in a target domain stems from "domain shift", i.e., the boundaries between the classes change in the embedding space which in turn increases possibility of misclassification. It has been shown that if an increased-margin classifier is trained in the source domain, it can generalize better than many methods that try to only align distributions (Tommasi and Caputo, 2013; Rostami, 2022). Inspired by this argument, our method is based on both aligning distributions in the embedding space and also inducing larger margins between classes by learning a "parametric distribution" for the source domain. Our idea is based on the empirical observation that when a deep network classifier is trained in a domain with annotated data, data points of classes form separable clusters in the embedding space modeled via the network responses in hidden layers. This means that the source distribution can be modeled as a multimodal distribution in the embedding space. Our work is based on using the multimodal distribution to induce larger margins between the class-specific clusters after an initial training phase in the source domain.

## 3 Cross-Domain Sentiment Analysis

Consider two sentiment analysis problems in a source domain $\mathcal{S}$ with an annotated dataset $D_\mathcal{S} = (\boldsymbol{X}_\mathcal{S}, \boldsymbol{Y}_\mathcal{S})$, where $\boldsymbol{X}_\mathcal{S} = [\boldsymbol{x}_1^s, \dots, \boldsymbol{x}_N^s] \in \mathcal{X} \subset \mathbb{R}^{d \times N}$ and $\boldsymbol{Y}_\mathcal{S} = [\boldsymbol{y}_1^s, ..., \boldsymbol{y}_N^s] \in \mathcal{Y} \subset \mathbb{R}^{k \times N}$, and

a target domain $\mathcal{T}$ with an unannotated dataset $D_\mathcal{T} = (\boldsymbol{X}_\mathcal{S})$, where $\boldsymbol{X}_\mathcal{T} = [\boldsymbol{x}_1^t, \dots, \boldsymbol{x}_N^t] \in \mathcal{X} \subset \mathbb{R}^{d \times M}$. The real-valued feature vectors $\boldsymbol{X}_\mathcal{S}$ and $\boldsymbol{X}_\mathcal{T}$ are obtained after pre-processing the input text data using common NLP methods, e.g., bag of words or word2vec. We consider that both domains share the same type of sentiments and hence the one-hot labels $\boldsymbol{y}_i^s$ encode $k$ sentiment types, e.g., negative or positive in binary sentiment analysis. We also assume that the source and the target feature data points are drawn independently and identically distributed from the domain-specific distributions $\boldsymbol{x}_i^s \sim p_S(\boldsymbol{x})$ and $\boldsymbol{x}_i^t \sim p_T(\boldsymbol{x})$, such that $p_T(\boldsymbol{x}) \neq p_S(\boldsymbol{x})$, i.e., there exists domain gap.

Given a family of parametric functions $f_\theta : \mathbb{R}^d \rightarrow \mathcal{Y}$, e.g., deep neural networks with learnable weights $\theta$, and considering an ideal labeling function $f(\cdot)$, e.g., $\forall(\boldsymbol{x}, \boldsymbol{y}) : \boldsymbol{y} = f(\boldsymbol{x})$, the goal is to search for the optimal predictor $f_{\theta^*}(\cdot)$ in this family for the target domain. This model should have minimal expected error, i.e., $\theta^* = \arg\min_\theta \{e_\theta\} = \arg\min_\theta \{\mathbb{E}_{\boldsymbol{x}^t \sim p_T(\boldsymbol{x})}(\mathcal{L}(f(\boldsymbol{x}^t), f_\theta(\boldsymbol{x}^t)))\}$, where $\mathcal{L}(\cdot)$ is a proper loss function and $\mathbb{E}(\cdot)$ denotes the expectation operator. Since the target domain data is unlabeled, the naive approach is to estimate the optimal model using the standard empirical risk minimization (ERM) in the source domain:

$$\begin{aligned} \hat{\theta} &= \arg\min_\theta \{\hat{e}_\theta(\boldsymbol{X}_\mathcal{S}, \boldsymbol{Y}_\mathcal{S}, \mathcal{L})\} \\ &= \arg\min_\theta \{\frac{1}{N} \sum_i \mathcal{L}(f_\theta(\boldsymbol{x}_i^s), \boldsymbol{y}_i^s)\}. \end{aligned} \tag{1}$$

Given a large enough labeled dataset in the source domain, ERM model generalizes well in the source domain. The source-trained model may also perform better than chance in a similar target domain, given cross-domain knowledge transfer, yet its performance will degrade in the target domain compared to its performance in the source domain because of existing distributional discrepancy between the two domains, $p_S \neq p_T$. Our goal is to benefit from the encoded information in the unlabeled target domain data points and adapt the source-trained classifier $f_{\hat{\theta}}$ to generalize better in the target domain. We use the common approach of reducing the domain gap across the two domains by mapping data points into a shared embedding.

We consider that the predictor model $f_\theta(\cdot)$ can be decomposed into a deep encoder subnetwork $\phi_{\boldsymbol{v}}(\cdot) : \mathcal{X} \rightarrow \mathcal{Z} \subset \mathbb{R}^p$ and a classifier subnetwork $h_{\boldsymbol{w}}(\cdot) : \mathcal{Z} \rightarrow \mathcal{Y}$ such that $f_\theta = h_{\boldsymbol{w}} \circ \phi_{\boldsymbol{v}}$, where

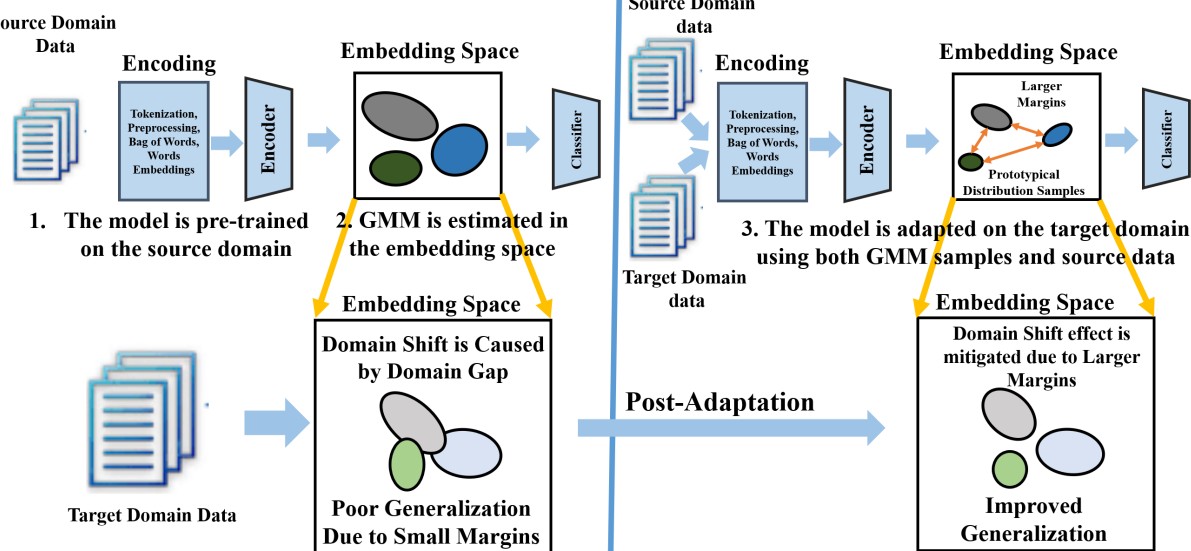

Figure 1: Architecture of the proposed cross-domain sentiment analysis framework. Left: separable data clusters are formed in the embedding space after initial supervised model training in the source domain and then following our previous work (Rostami, 2021), the internally learned multimodal distribution is estimated using a GMM. Right: random samples from the GMM with high-confident labels are drawn and then used to generate a pseudo-dataset. The pseudo-dataset helps to induce larger margins between the classes to mitigate domain shift in the target domain.

$\theta = (\boldsymbol{w}, \boldsymbol{v})$. Here, $\mathcal{Z}$ is an embedding space which is modeled by the encoder responses at its output. We assume that the classes have become separable for the source domain in this space after an initial training phase on the source domain (see Figure 1, left). If we can adapt the source-trained encoder network such that the two domains share similar distributions $\mathcal{Z}$, i.e., $\phi(p_{\mathcal{S}})(\cdot) \approx \phi(p_{\mathcal{T}})(\cdot)$, the embedding space would become domain-agnostic. As a result, the source-trained classifier network will generalize well in the target domain. A number of prior cross-domain sentiment analysis algorithms use this strategy, select a proper distribution metric to compute the distance between $\phi(p_{\mathcal{S}}(\boldsymbol{x}^s))$ and $\phi(p_{\mathcal{T}}(\boldsymbol{x}^t))$, and then train the encoder network to align the domains via minimizing it:

$$\hat{\boldsymbol{v}}, \hat{\boldsymbol{w}} = \arg \min_{\boldsymbol{v}, \boldsymbol{w}} \frac{1}{N} \sum_{i=1}^{N} \mathcal{L}\big(h_{\boldsymbol{w}}(\phi_{\boldsymbol{v}}(\boldsymbol{x}_i^s)), \boldsymbol{y}_i^s\big)$$
$$+ \lambda D\big(\phi_{\boldsymbol{v}}(p_{\mathcal{S}}(\boldsymbol{X}_{\mathcal{T}})), \phi_{\boldsymbol{v}}(p_{\mathcal{T}}(\boldsymbol{X}_{\mathcal{T}}))\big), \quad (2)$$

where $D(\cdot, \cdot)$ denotes a probability metric to measure the domain discrepancy and $\lambda$ is a trade-off parameter between the source ERM and the domain alignment term. We base our work on this general approach and use SWD (Lee et al., 2019) to compute $D(\cdot, \cdot)$ in (2). Using SWD has three advantages. First, SWD can be computed efficiently compared to WD based on a closed form solution of WD distance in 2D. Second, SWD can be com-

puted using the empirical samples that are drawn from the two distributions. Finally, SWD possesses a non-vanishing gradient even when the support of the two distributions do not overlap (Bonnotte, 2013; Lee et al., 2019). Hence SWD is suitable for solving deep learning problems which are normally handled using first-order gradient-based optimization techniques, e.g., Adam.For more background on SWD please refer to the Appendix (Section B).

While methods based on variations of Eq. (2) are effective to reduce the domain gap to some extent, our goal is to improve upon the baseline obtained by Eq. (2) by introducing a loss term that increases the margins between classes in the source domain (check the embedding space in Figure 1, right, for having a better intuition). By doing so, our goal is to mitigate the negative effect of domain shift.

## 4 Increasing Interclass Margins

Our idea for increasing margins between the classes is based on learning an intermediate parametric distribution in the embedding space. We demonstrate that this distribution can be used to induce larger margins between the classes. To this end, we consider that the classifier subnetwork consists of a softmax layer. This means that the classifier should become a maximum *a posteriori* (MAP) estimator after training to be able to assign a membership probability to a given input feature vec-

**Algorithm 1** $\text{SAIM}^2\ (\lambda, \tau)$

1: **Initial Training**:
2:   **Input:** source dataset $\mathcal{D}_{\mathcal{S}} = (\boldsymbol{X}_{\mathcal{S}}, \boldsymbol{Y}_{\mathcal{S}})$,
3:   **Training on the Source Domain:**
4:     $\hat{\theta}_0 = (\hat{\boldsymbol{w}}_0, \hat{\boldsymbol{v}}_0)$
5:     $= \arg\min_\theta \sum_i \mathcal{L}(f_\theta(\boldsymbol{x}_i^s), \boldsymbol{y}_i^s)$
6:   **Prototypical Distribution Estimation:**
7:     Use (4) and estimate $\alpha_j, \boldsymbol{\mu}_j, \Sigma_j$
8: **Model Adaptation**:
9:   **Input:** target dataset $\mathcal{D}_{\mathcal{T}} = \boldsymbol{X}_{\mathcal{T}}$
10:   **Pseudo-Dataset Generation:**
11:     $\hat{\mathcal{D}}_{\mathcal{P}} = (\mathbf{Z}_{\mathcal{P}}, \mathbf{Y}_{\mathcal{P}}) =$
12:     $([\boldsymbol{z}_1^p, \ldots, \boldsymbol{z}_N^p], [\boldsymbol{y}_1^p, \ldots, \boldsymbol{y}_N^p])$, where:
13:       $\boldsymbol{z}_i^p \sim \hat{p}_J(\boldsymbol{z}), 1 \leq i \leq N_p$
14:       $\boldsymbol{y}_i^p = \arg\max_j \{h_{\hat{\boldsymbol{w}}_0}(\boldsymbol{z}_i^p)\}$,
15:       $\max\{h_{\hat{\boldsymbol{w}}_0}(\boldsymbol{z}_i^p)\} > \tau$
16: **for** $itr = 1, \ldots, ITR$ **do**
17:   draw data batches from $\mathcal{D}_{\mathcal{S}}, \mathcal{D}_{\mathcal{T}}$, and $\mathcal{D}_{\mathcal{P}}$
18:   Update the model by solving (6)
19: **end for**

---

tor. Under this formulation, the model will generalize in the source domain if after supervised training of the model using the source data, the input distribution is transformed into a multi-modal distribution $p_J(\cdot) = \phi_{\boldsymbol{v}}(p_{\mathcal{S}})(\cdot)$ with $k$ modes in the embedding space (see Figure 1, left). Each mode of this distribution corresponds to one type of sentiments. The geometric distance between the modes of this distribution corresponds to the margins between classes. If we test the source-trained model in the target domain, the boundaries between class modes will change due to "domain shift", i.e., $\phi_{\boldsymbol{v}}(p_{\mathcal{T}})(\cdot) \neq \phi_{\boldsymbol{v}}(p_{\mathcal{S}})(\cdot)$. As visualized in Figure 1, if we increase the margins between the class-specific modes in the source domain, domain shift will likely cause less performance degradation (Tommasi and Caputo, 2013).

We estimate the multimodal distribution in the embedding as a Gaussian Mixture Model (GMM):

$$p_J(\boldsymbol{z}) = \sum_{j=1}^{k} \alpha_j \mathcal{N}(\boldsymbol{z}|\boldsymbol{\mu}_j, \boldsymbol{\Sigma}_j), \qquad (3)$$

where $\boldsymbol{\mu}_j$ and $\boldsymbol{\Sigma}_j$ denote the mean and co-variance matrices for each component and $\alpha_j$ denotes mixture weights for each component. We solve for these parameters to estimate the multimodal distribution. Note that unlike usual cases in which iterative and time-consuming algorithms such as expectation maximization algorithm need to be used

for estimating the GMM parameters, the source domain data points are labeled. As a result, we can estimate $\boldsymbol{\mu}_j$ and $\boldsymbol{\Sigma}_j$ for each component independently using standard MAP estimates. Similarly, the weights $\alpha_j$ can be computed by a MAP estimate. Let $\boldsymbol{S}_j$ denote the support set for class $j$ in the training dataset, i.e., $\boldsymbol{S}_j = \{(\boldsymbol{x}_i^s, \boldsymbol{y}_i^s) \in \mathcal{D}_{\mathcal{S}} | \arg\max \boldsymbol{y}_i^s = j\}$. To cancel out outliers, we include only those source samples in the $\boldsymbol{S}_j$ sets, for which the source-trained model predicts the corresponding labels correctly. The closed-form MAP estimate for the mode parameters is given as:

$$\hat{\alpha}_j = \frac{|\boldsymbol{S}_j|}{N}, \quad \hat{\boldsymbol{\mu}}_j = \sum_{(\boldsymbol{x}_i^s, \boldsymbol{y}_i^s) \in \boldsymbol{S}_j} \frac{1}{|\boldsymbol{S}_j|} \phi_v(\boldsymbol{x}_i^s),$$

$$\hat{\boldsymbol{\Sigma}}_j = \sum_{(\boldsymbol{x}_i^s, \boldsymbol{y}_i^s) \in \boldsymbol{S}_j} \frac{1}{|\boldsymbol{S}_j|} \big(\phi_v(\boldsymbol{x}_i^s) - \hat{\boldsymbol{\mu}}_j\big)^\top \big(\phi_v(\boldsymbol{x}_i^s) - \hat{\boldsymbol{\mu}}_j\big).$$

$$(4)$$

Computations in Eq. (4) can be done efficiently. For a complexity analysis, please refer to the Appendix (Section C). Our idea is to use this multimodal GMM distributional estimate to induce larger margins in the source domain (see Figure 1, right). We update the domain alignment term in (2) to induce larger margins. To this end, we augment the source domain samples in the domain alignment term with samples of a labeled pseudo-dataset $\mathcal{D}_{\mathcal{P}} = (\mathbf{Z}_{\mathcal{P}}, \mathbf{Y}_{\mathcal{P}})$ that we generate using the GMM estimate, where $\mathbf{Z}_{\mathcal{P}} = [\boldsymbol{z}_1^p, \ldots, \boldsymbol{z}_{N_p}^p] \in \mathbb{R}^{p \times N_p}, \mathbf{Y}_{\mathcal{P}} = [\boldsymbol{y}_1^p, \ldots, \boldsymbol{y}_{N_p}^p] \in \mathbb{R}^{k \times N_p}$. This pseudo-dataset is generated using the the GMM distribution. We draw samples from the GMM distributional estimate $\boldsymbol{z}_i^p \sim \hat{p}_J(\boldsymbol{z})$ for this purpose. To induce larger margins between classes, we feed the initial drawn samples into the classifier network and check the confidence level of the classifier about its predictions for these randomly drawn samples. We set a confidence threshold level $\tau \approx 1$ and only select a subset of samples for which the confidence level of the classifier is more than $\tau$:

$$(\boldsymbol{z}_i^p, \boldsymbol{y}_i^p) \in \mathcal{D}_{\mathcal{P}} \text{ if: } \boldsymbol{z}_i^p \sim \hat{p}_J(\boldsymbol{z}) \text{ and} \\ \max\{h(\boldsymbol{z}_i^p)\} > \tau \text{ and } \boldsymbol{y}_i^p = \arg\max_i\{h(\boldsymbol{z}_i^p)\}. \quad (5)$$

Given the GMM distributional form, selection of samples based on the threshold $\tau$ means that we include GMM samples that are closer to the class-specific mode means $\mu_i$ (see Figure 1). The margins between the clusters in the source domain increase if we use the generated pseudo-dataset for

domain alignment. Hence, we update Eq. (2) as:

$$\hat{\boldsymbol{v}}, \hat{\boldsymbol{w}} = \arg\min_{\boldsymbol{v}, \boldsymbol{w}} \Big\{ \frac{1}{N} \sum_{i=1}^{N} \mathcal{L}\big(h_{\boldsymbol{w}}(\phi_{\boldsymbol{w}}(\boldsymbol{x}_i^s)), \boldsymbol{y}_i^s\big)$$

$$+ \frac{1}{N_p} \sum_{i=1}^{N_p} \mathcal{L}\big(h_{\boldsymbol{w}}(\boldsymbol{z}_i^s), \boldsymbol{y}_i^s\big) + \lambda \hat{D}(\phi_{\boldsymbol{v}}(\boldsymbol{X}_{\mathcal{T}}), \boldsymbol{X}_{\mathcal{P}}))\Big)$$

$$+ \lambda \hat{D}(\phi_{\boldsymbol{v}}(\boldsymbol{X}_{\mathcal{S}}), \boldsymbol{X}_{\mathcal{P}})\Big\}, \qquad (6)$$

The first and the second terms in (6) are ERM terms for the source dataset and the generated pseudo-dataset to guarantee that the classifier continues to generalize well in the source domain after adaptation. The third and the fourth terms are empirical SWD losses (see Appendix for more details) that align the source and the target domain distributions using the pseudo-dataset which as we describe induces larger margins. The hope is that as visualized in Figure 1, these terms can reduce the effect of domain shift. Our proposed solution, named Sentiment Analysis using Increased-Margin Models (SAIM$^2$), is presented in Algorithm 1 and Figure 1.

## 5   Theoretical Analysis

Following a standard PAC-learning framework (Shalev-Shwartz and Ben-David, 2014), we prove that Algorithm 1 minimizes an upperbound for the target domain expected error. Consider that the PAC-learning hypothesis class to be the family of classifier sub-networks $\mathcal{H} = \{h_{\boldsymbol{w}}(\cdot) | h_{\boldsymbol{w}}(\cdot) : \mathcal{Z} \to \mathbb{R}^k, \boldsymbol{v} \in \mathbb{R}^V\}$, where $V$ denotes the number of learnable parameters. We represent the expected error for a model $h_{\boldsymbol{w}}(\cdot) \in \mathcal{H}$ on the source and the target domains by $e_{\mathcal{S}}(\boldsymbol{w})$ and $e_{\mathcal{T}}(\boldsymbol{w})$. Given the source and the target datasets, we can represent the empirical source and target distributions in the embedding space as $\hat{\mu}_{\mathcal{S}} = \frac{1}{N} \sum_{n=1}^{N} \delta(\phi_{\boldsymbol{v}}(\boldsymbol{x}_n^s))$ and $\hat{\mu}_{\mathcal{T}} = \frac{1}{M} \sum_{m=1}^{M} \delta(\phi_{\boldsymbol{v}}(\boldsymbol{x}_m^t))$. Similarly, we can build an empirical distribution for the multimodal distribution $\hat{\mu}_{\mathcal{P}} = \frac{1}{N_p} \sum_{q=1}^{N_p} \delta(\boldsymbol{z}_n^q)$. In our analysis we also use the notion of joint-optimal model $h_{\mathcal{S}, \mathcal{T}}(\cdot)$ which is defined as: $\boldsymbol{w}^* = \arg\min_{\boldsymbol{w}} e_{\mathcal{S}, \mathcal{T}} = \arg\min_{\boldsymbol{w}} \{e_{\mathcal{S}} + e_{\mathcal{T}}\}$ for any given domains $\mathcal{S}$ and $\mathcal{T}$. When we have labeled data in both domains, this is the best ERM-trained model. Existence of a good joint-trained model guarantees that the domains are related for positive transfer, e.g., similar sentiment polarities are encoded across the two domains.

**Theorem 1**: Consider that we use the procedure described in Algorithm 1 for cross-domain sentiment analysis, then the following inequality holds

for the target expected error:

$$e_{\mathcal{T}} \leq e_{\mathcal{S}} + \hat{D}(\hat{\mu}_{\mathcal{S}}, \hat{\mu}_{\mathcal{P}}) + \hat{D}(\hat{\mu}_{\mathcal{T}}, \hat{\mu}_{\mathcal{P}}) + (1 - \tau) + e_{\mathcal{S}, \mathcal{P}}$$

$$+ \sqrt{\big(2 \log(\frac{1}{\xi})/\zeta\big)} \Big(\sqrt{\frac{1}{N}} + \sqrt{\frac{1}{M}} + 2\sqrt{\frac{1}{N_p}}\Big), \qquad (7)$$

where $\xi$ is a constant depending on $\mathcal{L}(\cdot)$.

**Proof:** The complete proof is included in the Appendix (Section C).

Theorem 1 provides an explanation to justify Algorithm 1. We observe that all the terms in the upperbound of the target expected error in the right-hand side of (7) are minimized by $SAIM^2$. The source expected error is minimized as the first term in (6). The second and the third terms are minimized as the third and fourth terms of (6). The fourth term $1 - \tau$ will be small if we set $\tau \approx 1$. The term $e_{\mathcal{S}, \mathcal{P}}$ is minimized through the first and the second term of (6). This is highly important as using the pseudo-dataset provides a way to minimize this term. As can be seen in our proof in the Appendix, if we don't use the pseudo-dataset, this terms is replaced with $e_{\mathcal{S}, \mathcal{T}}$ which cannot be minimized directly due to lack of having annotated data in the target domain. The last term in (7) is a constant term that as common in PAC-learning can become negligible states that in order to train a good model if we have access to large enough datasets. Hence all the terms in the upperbound are minimized and if this upperbound is tight, then the process leads to training a good model for the target domain. If the two domain are related, e.g., share the same classes, and also classes become separable in the embedding space, i.e., GMM estimation error for the source domain distribution in the embedding space is small, then the upperbound is going to be likely tight. However, we highlight that the prospect of a tight upperbound is a condition for our algorithm to work in practical settings. Note, however, this is a common limitation for most parametric machine learning algorithms.

## 6   Experimental Validation

We have selected the most common setup for UDA sentiment analysis to perform our experiments for possibility of comparing our performance against prior works. Our implemented code is publicly available: https://github.com/digbose92/SAIM2

### 6.1   Experimental Setup

**Dataset and the cross-domain tasks:** Most existing works in cross-domain sentiment analysis re-

port performance on cross-domain tasks that are defined using the real-world Amazon Reviews benchmark dataset (Blitzer et al., 2007). The dataset is built using Amazon product reviews from four product domains: Books (B), DVD (D), Electronics (E), and Kitchen (K) appliances. Hence, 12 pairwise cross-domain tasks can be defined amongst the domains. Each review is considered to have positive (higher than 3 stars) or negative (3 stars or lower) sentiment. Each task consists of 2000 labeled reviews for the source domain and then 2000 unlabeled reviews for the target domain, and 2500–5500 examples for testing. We report our performance on the 12 definable cross-domain tasks. We report the average prediction accuracy and standard deviation (std) over 5 runs of our code.

**Preprocessing:** We have used two methods for backbone feature extraction: *tf-idf* and BERT features (Du et al., 2020). The most common setup in the literature is using *tf-idf* feature vector of bag-of-words unigrams and bigrams, where we encode each review as a $d = 5000$ dimensional or $d = 30000$ dimensional *tf-idf* feature vectors. Following (Du et al., 2020), qe also report performance when modern BERT features are used. Following the precedence, we first fine-tune a BERT backbone, followed by FCC layers, on the source domain to solve the sentiment analysis task. We call this baseline vanilla BERT (VBERT), analogues to SO for *tf-idf* features. Features extracted via VBERT are then used by the BERT-based methods for domain adaptation. Note that using BERT as an advanced text embedding method increases the absolute performance, i.e., pretraining of BERT on a large auxiliary dataset enables knowledge transfer. However, in model adaptation we must evaluate the algorithms according to the relative improvement of the model performance on the target domain compared to the solely source-trained model. Hence, only relative improvement over the baseline should be used for evaluating performance of domain adaptation algorithms.

**Methods for Comparison**: We compare our method against several existing algorithms that use *tf-idf* features. We compare against DSN (Bousmalis et al., 2016) CMD (Zellinger et al., 2017), ASYM (Saito et al., 2018), PBLM (Ziser and Reichart, 2018), MT-Tri (Ruder and Plank, 2018), TRL (Ziser and Reichart, 2019), and TAT (Liu et al., 2019). These works are representative of advances in the field based on various approaches.

DSN and CMD are similar to $SAIM^2$ in that both align distributions in an embedding space. DSN learns shared and domain-specific knowledge for each domain and aligns the shared knowledge using the mean-based maximum mean discrepancy metric. CMD uses the central moment discrepancy metric for domain alignment. ASYM benefits from the idea of pseudo-labeling of the target samples to updated the base model. MT-Tri is based on ASYM but it also benefits from multi-task learning. TRL and PBLM do not use distribution alignment and are based on the pivot based language model. TAT is a recent work that has used adversarial learning successfully for cross-domain sentiment analysis. All the methods except TAT that uses 30000 dimensional features, use 5000 dimensional features. Note that in the results, the methods are comparable if they use features with the same dimension for fair comparison. We report performance of the source only (SO) model as a lowerbound to demonstrate the effect of adapting the model over this baseline. We also compare against 4 BERT-based methods: vanilla fine-tuned BERT (VBERT), HATN (Li et al., 2018) which is based on a hierarchical attention transfer mechanism, adversarial training over VBERT (AT-BERT), and BERT-DA (Du et al., 2020). We provided results by the authors in our table. We report std if std is reported in the original paper.

**Model and optimization setup:** We used the feature-specific benchmark neural network architecture that is used in the above mentioned works for fair comparison. For *tf-idf*-based comparison, we used an encoder with one hidden dense layer with 50 nodes with sigmoid activation function. The classifiers consist of a softmax layer with two output nodes. We implemented our method in Keras, used adam optimizer and tuned the learning rate in the source domain. For BERT-based architecture, we have concatenated the BERT backbone with two fully connected layers with 256 and 2 nodes and ReLU and SoftMax nonlinear functions.

**Hyperparameter tuning:** An advantage for our algorithm is that there are only two major hyperparameters, $\tau$ and $\lambda$. We set $\tau = 0.99$ and $\lambda = 10^{-2}$. We tuned $\lambda$ based on a brute-force search. Note, however, we observed empirically that our algorithm is not sensitive to the value of $\lambda$. We used a GPU cluster equipped with 4 Nvidia Tesla P100-SXM2 GPUs. We used Keras for implementation[1].

---

[1]The code is available as a supplement.

| Features | Method | B→D | B→E | B→K | D→B | D→E | D→K |
|---|---|---|---|---|---|---|---|
| *tf-idf* $d = 5000$ | SO | 81.7 ± 0.2 | 74.0 ± 0.6 | 76.4 ± 1.0 | 74.5 ± 0.3 | 75.6 ± 0.7 | 79.5 ± 0.4 |
| | DSN | 82.8 ± 0.4 | 81.9 ± 0.5 | 84.4 ± 0.6 | 80.1 ± 1.3 | 81.4 ± 1.1 | 83.3 ± 0.7 |
| | CMD | 82.6 ± 0.3 | 81.5 ± 0.6 | 84.4 ± 0.3 | **80.7 ± 0.6** | 82.2 ± 0.5 | 84.8 ± 0.2 |
| | ASYM | 80.7 | 79.8 | 82.5 | 73.2 | 77.0 | 82.5 |
| | PBLM | 84.2 | 77.6 | 82.5 | 82.5 | 79.6 | 83.2 |
| | MT-Tri | 81.2 | 78.0 | 78.8 | 77.1 | 81.0 | 79.5 |
| | TRL | 82.2 | - | 82.7 | - | - | - |
| | $SAIM^2$ | **83.2 ± 0.2** | **83.9 ± 0.3** | **85.9 ± 0.3** | 80.3 ± 0.4 | **84.2 ± 0.3** | **87.3 ± 0.2** |
| *tf-idf* $d = 30000$ | TAT | 84.5 | 80.1 | 83.6 | 81.9 | 81.9 | 84.0 |
| | $SAIM^2$ | **86.2 ± 0.2** | **85.1 ± 0.2** | **87.6 ± 0.2** | 80.9 ± 0.5 | **85.2 ± 0.2** | **88.6 ± 0.2** |
| BERT | VBERT | 89.0 | 88.0 | 89.1 | 89.4 | 86.6 | 87.5 |
| | VBERT* | 85.3 | 85.8 | 88.8 | 85.0 | 85.8 | 87.0 |
| | HATN | 89.4 | 87.2 | 89.4 | 89.8 | 87.0 | 87.6 |
| | AT-BERT | 89.7 | 87.3 | 89.6 | 89.6 | 86.1 | 87.7 |
| | BERT-DA | **89.8** | 88.1 | **90.7** | 90.4 | **88.1** | **88.6** |
| | $SAIM^2$ | 87.5 | **88.3** | 88.0 | **90.5** | 87.3 | 88.5 |

| Features | Method | E→B | E→D | E→K | K→B | K→D | K→E |
|---|---|---|---|---|---|---|---|
| *tf-idf* $d = 5000$ | SO | 72.3 ± 1.5 | 74.2 ± 0.6 | 85.6 ± 0.6 | 73.1 ± 0.1 | 75.2 ± 0.7 | 85.4 ± 1.0 |
| | DSN | 75.1 ± 0.4 | 77.1 ± 0.3 | 87.2 ± 0.7 | 76.4 ± 0.5 | 78.0 ± 1.4 | 86.7 ± 0.7 |
| | CMD | 74.9 ± 0.6 | 77.4 ± 0.3 | 86.4 ± 0.9 | 75.8 ± 0.3 | 77.7 ± 0.4 | 86.7 ± 0.6 |
| | ASYM | 73.2 | 72.9 | 86.9 | 72.5 | 74.9 | 84.6 |
| | PBLM | 71.4 | 75.0 | 87.8 | 74.2 | **79.8** | **87.1** |
| | MT-Tri | 73.5 | 75.4 | 87.2 | 73.8 | 77.8 | 86.0 |
| | TRL | - | 75.8 | - | 72.1 | - | - |
| | $SAIM^2$ | **78.6 ± 0.4** | **79.7 ± 0.2** | **89.2 ± 0.2** | **76.7 ± 0.4** | 79.1 ± 0.4 | 87.0 ± 0.1 |
| *tf-idf* $d = 3000$ | TAT | **83.2** | 77.9 | 90.0 | 75.8 | 77.7 | **88.2** |
| | $SAIM^2$ | 78.8 ± 0.3 | **78.9 ± 0.3** | **90.1 ± 0.2** | **78.1 ± 0.2** | **78.8 ± 0.4** | 88.1 ± 0.1 |
| BERT | VBERT | 86.5 | 86.2 | 91.6 | 87.6 | 87.3 | 90.5 |
| | VBERT* | 84.3 | 78.8 | 86.0 | 84.5 | 81.0 | 87.0 |
| | HATN | 87.2 | 88.8 | 92.0 | 87.9 | 87.9 | 90.3 |
| | AT-BERT | 87.2 | 88.2 | 91.9 | 87.7 | 87.7 | 90.3 |
| | BERT-DA | 88.3 | **89.0** | **92.8** | 87.9 | **88.4** | 90.6 |
| | $SAIM^2$ | **89.0** | 85.5 | 90.8 | **88.0** | 84.5 | **91.3** |

Table 1: Classification accuracy for the cross-domain sentiment analysis tasks for the Amazon Reviews dataset. Bold font denotes the method with maximum performance in each column for the corresponding in0ut feature.

## 6.2 Comparative Results

Our results are reported in Table 1. In this table, bold font denotes the best performance among the methods that use the same feature type. For the case of $tf - idf$ with $d = 5000$ which is the classic setting, we see that $SAIM^2$ algorithm performs reasonably well and in most cases leads to the best performance. Note that this is not unexpected as none of the methods has the best performance across all tasks. We observe from this table that overall the methods DSN and CMD which are based on aligning the source and target distributions- which are more similar to our algorithm- have relatively similar performances. This observation suggests that we should not expect considerable performance boost if we simply align the distributions by designing a new alignment loss function. This means that outperformance of $SAIM^2$ compared to these methods stems from inducing larger margins. We verify this intuition in our ablative study. We also observe that increasing the dimension of *tf-idf* features to 30000 leads to performance boost which is probably the reason behind good performance of TAT. We conclude that for a fair comparison on

these tasks, we should always use the same dimension to generate the features.

For the BERT-based methods, we have reported two versions of vanilla BERT. VBERT denotes results reported by (Du et al., 2020) and VBERT* denotes the results that we could obtain following all the guidelines that are provided in (Du et al., 2020). We first note that vanilla BERT performance are considerably higher than SO *tf-idf*-based performance. This significant boost is likely due to the pretraining of BERT on an external large corpus and should not be considered a reason for improved cross-domain performance. We also note that our performance is competitive for many tasks. We note that lesser performance of our method in some cases stems from smaller VBERT* performance versus VBERT. We speculate our VBERT performance can be boosted to a performance similar to VBERT* through better optimization hyperparameter tuning. However, if we base our comparison on the relative improved performance that is gained as the result of the adaptation procedure, our model is highly competitive. As also predictable from our theoretical result, we empirically conclude that the initial source-only performance of the model on the target domain is extremely important.

To provide an empirical exploration to validate the intuition we used for our rationale, we have used UMAP (McInnes et al., 2018) to reduce the dimension of the data representations in the 50D embedding space to two for the purpose of 2D visualization. Figure 2 visualizes the testing splits of the source domain before model adaptation, the testing splits of the target domain before and after model adaptation, and finally random samples drawn from the estimated GMM distribution for the D→K task. Each point in the figure represents one data point and each color represents one of the sentiments. Observing Figure 2a and Figure 2b, we conclude that the estimated GMM distribution approximates the source domain distribution reasonably well and at the same time, a margin between the classes in the boundary region is observable. Figure 2c visualizes the target domain samples prior to model adaptation. As expected, we observe that domain gap has caused less separations between the clusters for the classes, as also evident from SO performance in Table 1. Figure 2d visualizes the target domain samples after adaptation using $SAIM^2$ algorithm. Comparing Figure 2d with Figure 2c and Figure 2a, we see

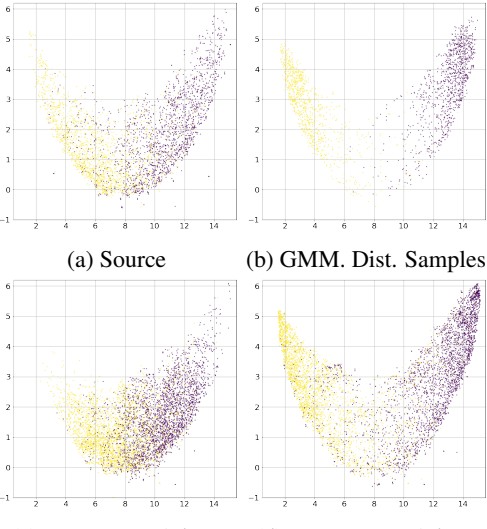

| (a) Source | (b) GMM. Dist. Samples |
| (c) Target: Pre-Adapt. | (d) Target: Post-Adapt. |

Figure 2: UMAP visualization for the D→K task: (a) the source domain testing split, (b) the GMM distribution samples, (c) the target domain testing split pre- and (d) post-adaptation. (Best viewed in color).

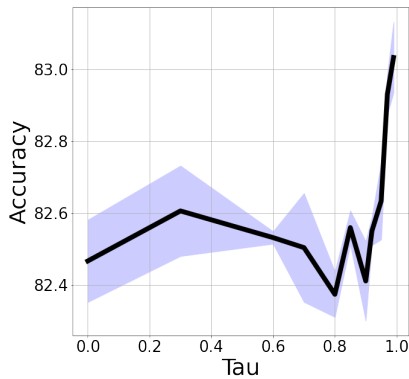

Figure 3: Effect of $\tau$ on performance.

that the classes have become more separated. Also, careful comparison of Figure 2d and Figure 2b reveals $SAIM^2$ algorithm has led to a bias in the target domain to move the data points further away from the boundary. These visualizations serve as an empirical verification for our theoretical analysis.

### 6.3 Ablation Studies

First note that the source only (SO) model and vanilla BERT results, which are trained using (1), already serve as a basic ablative experiment to study the effect of domain alignment. Observed improvements over this baseline demonstrate positive effect of domain adaptation on performance. The amount of relative improvement also measures the quality of model adaptation algorithms. Additionally, we provide two sets of ablative experiments.

In Table 2, we have provided an additional abla-

Table 2: Performance for AO.

| Task | B→D | B→E | B→K |
|------|-----|-----|-----|
|  | 81.9 ± 0.5 | 80.9 ± 0.8 | 83.2 ± 0.8 |
| Task | D→B | D→E | D→K |
|  | 74.0 ± 0.9 | 80.9 ± 0.7 | 83.4 ± 0.6 |
| Task | E→B | E→D | E→K |
|  | 74.1 ± 0.6 | 74.0 ± 0.3 | 87.8 ± 0.9 |
| Task | D→B | D→E | D→K |
|  | 74.1 ± 0.6 | 74.0 ± 0.3 | 87.8 ± 0.9 |

tive experiment to study the effect of the increased interclass margin on performance. We have reported result of alignment only (AO) model adaptation based on Eq. (2). The AO model does not benefit from the margins that $SAIM^2$ algorithm induces between the classes via solving Eq. (6). Comparing AO results with Table 1, we can conclude that the effect of increased interclass margins is important in our final performance. We can clearly see that compared to other cross-domain sentiment analysis methods, the performance boost for our algorithm stems from inducing large margins. This suggests that researchers may need to investigate secondary mechanism for improving domain adaptation in NLP domains, in addition to probability distribution alignment.

We have also studied the effect of the value of the confidence parameter on performance. In Figure 3, we have visualized the performance of our algorithm for the task $B \rightarrow D$ when $\tau$ is varied in the interval $[0, 0.99]$, where the solid black line shows average performances and the shaded blue region shows the standard deviation. When $\tau = 0$, the samples are not necessarily confident samples. We observe that as we increase the value of $\tau$ towards 1, the performance increases as a result of inducing larger interclass margins. For values $\tau > 0.8$, the performance has less variance which suggests robustness of performance if $\tau \approx 1$. These empirical observations about $\tau$ values validate our theoretical result as stated in the upperbound (7).

Finally, our empirical exploration demonstrates that our algorithm is robust with respect to data imbalance which is an important advantage in domain adaptation application. Due to the space limit, this study is included in the Appendix (Section E).

## 7 Conclusions

We developed a method for cross-domain sentiment analysis based on aligning two domain-specific dis-

tributions in a shared embedding space and inducing larger margins between the classes using an intermediate multi-modal GMM distribution. We theoretically demonstrated that our approach minimizes an upperbound for the target domain error. Our experiments demonstrate that our algorithm is effective and compares favorably against the state-of-the-art. A future research direction is to address cross-domain sentiment analysis when distributed source domains exist (Stan and Rostami, 2022).

## Limitations

Our approach is based on aligning the distribution by assuming that initially the two distributions have nontaxable overlap. However, if the domain gap is initially large, our approach may fail to align the distributions class-conditionally. As a result, some classes might be aligned incorrectly, despite the fact that the two domains end up having similar distributions. To resolve this limitations, we need to use techniques such as pseudo-labeling methods.

Furthermore, our approach operates under the assumption that the source domain data is readily accessible and can be directly utilized during the adaptation phase. However, there are situations where the source data is private and cannot be shared. In order to address this issue, it becomes crucial for us to devise a mechanism that can effectively align the distributions of the source and target domains without relying on direct usage of the source domain data. To tackle this challenge, we need to explore alternative methods that can indirectly capture the relevant information from the source domain without compromising its privacy. This requires developing innovative techniques that leverage shared latent representations or intermediate feature spaces for domain alignment. By establishing a connection between the source and target domains at a higher level, we may be able to align their distributions and bridge the gap between them without directly sharing data.

Finally, as large language models (LLMs) are being increasingly adopted in many applications, the effect of domain gap can be mitigated because LLMs have been pre-trained on extremely large corpus that contain a large amount of information. As a result, they are applicable to many domains without being affected by domain shift. As a result, UDA may not be as significant challge as it used to be, but still if domain shift exists in a target domain, using UDA is going to be very beneficial.

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

## A Sliced Wasserstein distance

We relied on minimizing the Sliced Wasserstein (SWD) distance for domain alignment. SWD is defined based on the Wasserstein distance (WD) and is a mean to come up with a more computationally efficient distribution metric. The WD between two distributions $p_\mathcal{S}$ and $p_\mathcal{T}$, is defined as:

$$W_c(p_\mathcal{S}, p_\mathcal{T}) = \inf_{\gamma \in \Gamma(p_\mathcal{S}, p_\mathcal{T})} \int_{X \times Y} c(x, y) d\gamma(x, y) \quad (8)$$

where $\Gamma(p_\mathcal{S}, p_\mathcal{T})$ is the set of all joint distributions $p_{\mathcal{S},\mathcal{T}}$ with marginal single variable distributions $p_\mathcal{S}$ and $p_\mathcal{T}$, and $c : X \times Y \to \mathbb{R}^+$ is the cost function, e.g., $\ell_2$-norm Euclidean distance.

We observe that computing WD involves solving a complicated optimization problem in the general case. However, when the two distributions are $1-$dimensional, WD has a closed-form solution:

$$W_c(p_\mathcal{S}, p_\mathcal{T}) = \int_0^1 c(P_\mathcal{S}^{-1}(\tau), P_\mathcal{T}^{-1}(\tau)) d\tau, \quad (9)$$

where $P_\mathcal{S}$ and $P_\mathcal{T}$ are the cumulative distributions of the 1D distributions $p_\mathcal{S}$ and $p_\mathcal{T}$. This closed-form solution motivates the definition of SWD in order to extend applicability of (9) for higher dimensional distributions.

SWD is defined based on the idea of slice sampling (Neal, 2003). The idea is to project two $d$-dimensional distributions into their marginal one-dimensional distributions along a subspace, i.e., slicing the high-dimensional distributions, and then compute the distance between the two distribution by integrating over all the WD between the resulting 1D marginal probability distributions over all possible 1D subspaces using the closed form solution of WD. This can be a good replacement for the WD as any probability distribution can be expressed by the set of $1-$dimensional marginal projection distributions (Helgason, 2011). More specifically, a one-dimensional slice of the distribution for the distribution $p_\mathcal{S}$ is defined:

$$\mathcal{R}p_\mathcal{S}(t; \boldsymbol{\gamma}) = \int_{\mathcal{S}^{d-1}} p_\mathcal{S}(\boldsymbol{x}) \boldsymbol{\delta}(t - \langle \boldsymbol{\gamma}, \boldsymbol{x} \rangle) d\boldsymbol{x}, \quad (10)$$

where $\boldsymbol{\delta}(\cdot)$ denotes the Kronecker delta function, $\langle \cdot, \cdot \rangle$ denotes the vector inner dot product, $\mathbb{S}^{d-1}$ is the $d$-dimensional unit sphere, and $\boldsymbol{\gamma}$ is the projection direction.

The SWD is defined as the integral of all WD between the sliced distributions over all 1D subspaces

$\boldsymbol{\gamma}$ on the unit sphere as follows:

$$SW(p_\mathcal{S}, p_\mathcal{T}) = \int_{\mathbb{S}^{d-1}} W(\mathcal{R}p_\mathcal{S}(\cdot; \gamma), \mathcal{R}p_\mathcal{T}(\cdot; \gamma)) d\gamma \quad (11)$$

The main advantage of using SWD is that, computing SWD does not require solving a numerically expensive optimization.

In our practical setting, only samples from the distributions are available and we don't have the distributional form. Another advantage of SWD is that its empirical version can be computed based on the one-dimensional empirical WD. One-dimensional empirical WD be approximated as the $\ell_p$-distance between the sorted samples. We can compute merely the integrand function in (11) for a known $\gamma$ and then the integral in (11) via Monte Carlo style numerical integration. To this end, we draw random projection subspace $\boldsymbol{\gamma}$ from a uniform distribution that is defined over the unit sphere and then compute 1D WD along this sample. We can then approximate the integral in (11) by computing the arithmetic average over a suitably large enough number of drawn samples. More specifically, the SWD between $f$-dimensional samples $\{\phi(\boldsymbol{x}_i^\mathcal{S}) \in \mathbb{R}^f \sim p_\mathcal{S}\}_{i=1}^M$ and $\{\phi(\boldsymbol{x}_i^\mathcal{T}) \in \mathbb{R}^f \sim p_\mathcal{T}\}_{j=1}^M$ in our setting can be approximated as the following sum:

$$SW^2(p_\mathcal{S}, p_\mathcal{T}) \approx$$
$$\frac{1}{L} \sum_{l=1}^L \sum_{i=1}^M |\langle \gamma_l, \phi(\boldsymbol{x}_{s_l[i]}^\mathcal{S}) \rangle - \langle \gamma_l, \phi(\boldsymbol{x}_{t_l[i]}^\mathcal{T}) \rangle|^2 \quad (12)$$

where $\gamma_l \in \mathbb{S}^{f-1}$ is uniformly drawn random sample from the unit $f$-dimensional ball $\mathbb{S}^{f-1}$, and $s_l[i]$ and $t_l[i]$ are the sorted indices of $\{\gamma_l \cdot \phi(\boldsymbol{x}_i)\}_{i=1}^M$ for source and target domains, respectively.

We utilize this empirical version of SWD in (12) to align the distributions in the embedding space. Note that the function in (12) is differentiable with respect to the encoder parameters and hence we can use gradient-based optimization techniques to minimize it with respect to the model parameters.

## B Proof of Theorem 1

We use the following theorem by Redko et al. (Redko and Sebban, 2017) and a result by Bolley (Bolley et al., 2007) on convergence of the empirical distribution to the true distribution in terms of the WD distance in our proof.

**Theorem 2 (Redko et al. (Redko and Sebban, 2017)):** Under the assumptions described in our

framework, assume that a model is trained on the source domain, then for any $d' > d$ and $\zeta < \sqrt{2}$, there exists a constant number $N_0$ depending on $d'$ such that for any $\boldsymbol{x}_i > 0$ and $\min(N, M) \geq \max(\boldsymbol{x}i^{-(d'+2),1})$ with probability at least $1 - \boldsymbol{x}_i$, the following holds:

$$e_{\mathcal{T}} \leq e_{\mathcal{S}} + W(\hat{\mu}_{\mathcal{T}}, \hat{\mu}_{\mathcal{S}}) + e_{\mathcal{S},\mathcal{T}} \\ + \sqrt{\left(2\log(\frac{1}{\boldsymbol{x}_i})/\zeta\right)}\left(\sqrt{\frac{1}{N}} + \sqrt{\frac{1}{M}}\right). \quad (13)$$

Theorem 2 provides an upperbound for the performance of a source-trained model in the target domain Redko et al. (Redko and Sebban, 2017) prove Theorem 2 for a binary classification setting. We also provide our proof in this case but it can be extended.

The second term in Eq. (13) demonstrates the effect of domain shift on the performance of a source-trained model in a target domain. When the distance between the two distributions is significant, this term will be large and hence the upperbound in Eq. (13) will be loose which means potential performance degradation. Our algorithm mitigates domain gap because this term is minimized by minimization of the second and the third terms in Theorem 1.

**Theorem 1** : Consider that we use the procedure described in Algorithm 1 for cross-domain sentiment analysis, then the following inequality holds for the target expected error:

$$e_{\mathcal{T}} \leq e_{\mathcal{S}} + \hat{D}(\hat{\mu}_{\mathcal{S}}, \hat{\mu}_{\mathcal{P}}) + \hat{D}(\hat{\mu}_{\mathcal{T}}, \hat{\mu}_{\mathcal{P}}) + (1 - \tau) \\ + e_{\mathcal{S},\mathcal{P}} + \sqrt{\left(2\log\frac{1}{\boldsymbol{x}_i})/\zeta\right)}\left(\sqrt{\frac{1}{N}} + \sqrt{\frac{1}{M}} + 2\sqrt{\frac{1}{N_p}}\right), \quad (14)$$

where $\boldsymbol{x}_i$ is a constant which depends on $\mathcal{L}(\cdot)$ and $e'_C(\boldsymbol{w}^*)$ denotes the expected risk of the optimally joint trained model when used on both the source domain and the pseudo-dataset.

**Proof:** Due to the construction of the pseudo-dataset, the probability that the predicted labels for the pseudo-data points to be false is equal to $1 - \tau$. Let:

$$|\mathcal{L}(h_{\boldsymbol{w}_0}(\boldsymbol{z}_i^p), \boldsymbol{y}_i^p) - \mathcal{L}(h_{\boldsymbol{w}_0}(\boldsymbol{z}_i^p), \hat{\boldsymbol{y}}_i^p)| = \\ \begin{cases} 0, & \text{if } \boldsymbol{y}_i^t = \hat{\boldsymbol{y}}_i^t. \\ 1, & \text{otherwise.} \end{cases} \quad (15)$$

We use Jensen's inequality and take expectation on both sides of (15) to deduce:

$$|e_{\mathcal{P}} - e_{\mathcal{T}}| \leq \\ \mathbb{E}\left(|\mathcal{L}(h_{\boldsymbol{w}_0}(\boldsymbol{z}_i^p), \boldsymbol{y}_i^p) - \mathcal{L}(h_{\boldsymbol{w}_0}(\boldsymbol{z}_i^p), \hat{\boldsymbol{y}}_i^p)|\right) \quad (16) \\ \leq (1 - \tau).$$

Applying (16) in the below, deduce:

$$e_{\mathcal{S}} + e_{\mathcal{T}} = e_{\mathcal{S}} + e_{\mathcal{T}} + e_{\mathcal{P}} - e_{\mathcal{P}} \leq \\ e_{\mathcal{S}} + e_{\mathcal{P}} + |e_{\mathcal{T}} - e_{\mathcal{P}}| \leq e_{\mathcal{S}} + e_{\mathcal{P}} + (1 - \tau). \quad (17)$$

Taking infimum on both sides of (17), we deduce:

$$e_{\mathcal{S},\mathcal{T}} \leq e_{\mathcal{S},\mathcal{P}} + (1 - \tau). \quad (18)$$

Now by considering Theorem 2 for the two domains $\mathcal{S}$ and $\mathcal{T}$ and then using (18) in (13), we can conclude:

$$e_{\mathcal{T}} \leq e_{\mathcal{S}} + D(\hat{\mu}_{\mathcal{T}}, \hat{\mu}_{\mathcal{S}}) + e_{\mathcal{S},\mathcal{P}} + (1 - \tau) \\ + \sqrt{\left(2\log(\frac{1}{\boldsymbol{x}_i})/\zeta\right)}\left(\sqrt{\frac{1}{N}} + \sqrt{\frac{1}{M}}\right). \quad (19)$$

Now using the triangular inequality on the metrics we can deduce:

$$D(\hat{\mu}_{\mathcal{T}}, \hat{\mu}_{\mathcal{S}}) \leq D(\hat{\mu}_{\mathcal{T}}, \mu_{\mathcal{P}}) + D(\hat{\mu}_{\mathcal{S}}, \mu_{\mathcal{P}}) \\ \leq D(\hat{\mu}_{\mathcal{T}}, \hat{\mu}_{\mathcal{P}}) + D(\hat{\mu}_{\mathcal{S}}, \hat{\mu}_{\mathcal{P}}) + 2D(\hat{\mu}_{\mathcal{P}}, \mu_{\mathcal{P}}). \quad (20)$$

Now we replace the term $D(\hat{\mu}_{\mathcal{P}}, \mu_{\mathcal{P}})$ with its empirical counterpart using Theorem 1.1 in the work by (Bolley et al., 2007).

**Theorem 3** (Theorem 1.1 by Bolley et al. (Bolley et al., 2007)): consider that $p(\cdot) \in \mathcal{P}(\mathcal{Z})$ and $\int_{\mathcal{Z}} \exp\left(\alpha\|\boldsymbol{x}\|_2^2\right)dp(\boldsymbol{x}) < \infty$ for some $\alpha > 0$. Let $\hat{p}(\boldsymbol{x}) = \frac{1}{N}\sum_i \delta(\boldsymbol{x}_i)$ denote the empirical distribution that is built from the samples $\{\boldsymbol{x}_i\}_{i=1}^N$ that are drawn i.i.d from $\boldsymbol{x}_i \sim p(\boldsymbol{x})$. Then for any $d' > d$ and $\boldsymbol{x}_i < \sqrt{2}$, there exists $N_0$ such that for any $\epsilon > 0$ and $N \geq N_o \max(1, \epsilon^{-(d'+2)})$, we have:

$$P(W(p, \hat{p}) > \epsilon) \leq \exp(-\frac{-\boldsymbol{x}_i}{2}N\epsilon^2), \quad (21)$$

where $W$ denotes the WD distance. This relation measures the distance between the empirical distribution and the true distribution, expressed in the WD distance.

Applying (20) and (21) on (19) concludes Theorem 2 as stated:

$$e_{\mathcal{T}} \leq e_{\mathcal{S}} + D(\hat{\mu}_{\mathcal{S}}, \hat{\mu}_{\mathcal{P}}) + D(\hat{\mu}_{\mathcal{T}}, \hat{\mu}_{\mathcal{P}}) + (1 - \tau) + e_{\mathcal{S},\mathcal{P}} + \\ \sqrt{\left(2\log(\frac{1}{\boldsymbol{x}_i})/\zeta\right)}\left(\sqrt{\frac{1}{N}} + \sqrt{\frac{1}{M}} + 2\sqrt{\frac{1}{N_p}}\right), \quad (22)$$

| Task | B→D | B→E | B→K | D→B | D→E | D→K |
|------|-----|-----|-----|-----|-----|-----|
| 80/20 | 82.8 ± 0.3 | 83.2 ± 0.5 | 85.5 ± 0.3 | 78.7 ± 0.2 | 83.3 ± 0.2 | 86.8 ± 0.2 |
| 90/10 | 82.9 ± 0.5 | 83.4 ± 0.3 | 85.8 ± 0.2 | 78.5 ± 0.4 | 83.3 ± 0.4 | 86.8 ± 0.3 |
| Task | E→B | E→D | E→K | K→B | K→D | K→E |
| 80/20 | 78.7 ± 0.2 | 78.5 ± 0.5 | 88.6 ± 0.1 | 76.3 ± 0.6 | 77.9 ± 0.4 | 86.6 ± 0.1 |
| 90/10 | 78.7 ± 0.2 | 78.0 ± 0.4 | 88.0 ± 0.2 | 76.5 ± 0.5 | 77.3 ± 0.3 | 86.7 ± 0.2 |

Table 3: Effect of label-imbalance on performance.

## C Complexity analysis for GMM estimation

Estimating a GMM distribution usually is a computationally expensive tasks. The major reason is that normally the data points are unlabeled. This would necessitate relying on iterative algorithms such expectation maximization (EM) algorithm (Moon, 1996). Preforming iterative E and M steps until convergence leads to high computational complexity (Roweis, 1998). However, estimating the multimodal distribution with a GMM distribution is much simpler in our learning setting. Existence of labels helps us to decouple the Gaussian components and compute the parameters using MAP estimate for each of the mode parameters in one step as follows:

$$
\begin{aligned}
\hat{\alpha}_j &= \frac{|\boldsymbol{S}_j|}{N}, \\
\hat{\boldsymbol{\mu}}_j &= \sum_{(\boldsymbol{x}_i^s, \boldsymbol{y}_i^s) \in \boldsymbol{S}_j} \frac{1}{|\boldsymbol{S}_j|} \phi_v(\boldsymbol{x}_i^s), \\
\hat{\boldsymbol{\Sigma}}_j &= \sum_{(\boldsymbol{x}_i^s, \boldsymbol{y}_i^s) \in \boldsymbol{S}_j} \frac{1}{|\boldsymbol{S}_j|} \big( \phi_v(\boldsymbol{x}_i^s) - \hat{\boldsymbol{\mu}}_j \big)^\top \big( \phi_v(\boldsymbol{x}_i^s) - \hat{\boldsymbol{\mu}}_j \big).
\end{aligned}
\tag{23}
$$

Given the above and considering that the source domain data is balanced, complexity of computing $\alpha_j$ is $O(N)$ (just checking whether data points $\boldsymbol{x}_i^s$ belong to $\boldsymbol{S}_j$). Complexity of computing $\boldsymbol{\mu}_j$ is $O(NF/k)$, where $F$ is the dimension of the embedding space. Complexity of computing the co-variance matrices $\boldsymbol{\Sigma}_j$ is $O(F(\frac{N}{k})^2)$. Since, we have $k$ components, the total complexity of computing GMM is $O(\frac{FN^2}{k})$. If $O(F) \approx O(k)$, which seems to be a reasonable practical assumption, then the total complexity of computing GMM would be $O(N^2)$. Given the large number of learnable parameters in most deep neural networks which are more than $N$ for most cases, this complexity is fully dominated by complexity of a single step of backpropagation. Hence, this computing the GMM parameters does not increase the computational complexity for.

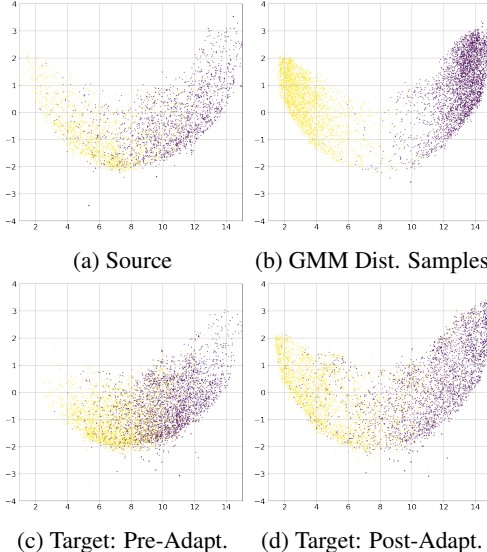

(a) Source      (b) GMM Dist. Samples

(c) Target: Pre-Adapt.      (d) Target: Post-Adapt.

Figure 4: UMAP visualization for the task D→K task in the imbalanced regime of $90\%/10\%$: (a) the source domain testing split, (b) the GMM distribution samples, (c) the target domain testing split pre-, and (d) post-domain adaptation. (Best viewed in color).

## D Effect of Data Imbalance on Performance

In practical settings of domain adaptation, the label distribution for the target domain training dataset cannot be enforced to be balanced due to the absence of labels. To study the effect of label imbalance on the performance of the proposed algorithm, we synthetically design imbalanced target domain datasets using the Amazon Reviews dataset. We designed two experiments, where the target domain datasets have the 90%/10% and 80%/20% ratios of imbalance between the positive and negative classes, respectively. We have provided domain adaptation results using these two imbalanced scenarios in Table 3 using *tf-idf* features with $d = 5000$. Comparing Table 3 with Table 1, we observe that performance of our algorithm has slightly degraded to relatively similar values for both scenarios. This degradation is expected because the majority class swamps the

minority class when generating batches for optimization. Note, however, our algorithm has been robust to a large extent with respect to label imbalance which demonstrates its practical suitability, when balanced datasets cannot be guaranteed.

For an intuitive sanity check, we have presented the UMAP visualization for the testing split of the task D→K for the scenario 90%/10% in Figure 4. Observations in Figure 4 match what we reported in Table 3, confirming that our algorithm can increase interclass margins when the target domain dataset is imbalanced but dominance of the majority class has led to less separation between the classes for the tasks built using the Amazon reviews dataset.