# OpenReview forum: "Domain Adaptation for Sentiment Analysis Using Robust Internal Representations"
_EMNLP/2023/Conference — EMNLP 2023 Findings_

### Official Review · Reviewer_Nxo2 · 2023-08-02

**Typos Grammar Style And Presentation Improvements:** 1. Some captions of certain tables li…
**Soundness:** 3

**Excitement:**

4: Strong: This paper deepens the understanding of some phenomenon or lowers the barriers to an existing research direction.

**Paper Topic And Main Contributions:**

This paper proposes a method for cross-domain sentiment analysis based on aligning two domain-specific distributions in a shared embedding space and inducing larger margins between the classes using an intermediate multi-modal GMM distribution to mitigate the "domain gap" between different applications.
The contribution is reflected in that learn a parametric distribution for the source domain in a cross-domain embedding space by estimating  the distribution as a Gaussian mixture model (GMM) and then use the GMM distribution to align the source and the target distributions using confident samples, drawn for them GMM to increase the interclass margins to reduce the domain gap.
This paper provides theoretical and empirical analysis for the effectiveness of the proposal.

**Reasons To Accept:**

1. The idea is interesting, in which it uses Gaussian mixture model (GMM) to align different domain distributions and generates pseudo-dataset for a larger margin between the clusters in the source domain.
2. This paper provides theoretical and empirical analysis for the effectiveness of the proposal.
3. The proposed method shows promising improvement over the chosen baselines, provided that the results are reproducible when the code is released.

**Reasons To Reject:**

The proposed method is mainly effective when using the traditional text features like tf-idf, by mitigating the domain gap. But when employing the pretrained language model like BERT and large language model (LLM), which are pretrained in large scale or extremely large corpus and performs strong domain transfer capability, this proposed method may be less competitive compared to them. The experiments in the paper prove the limitation.

**Reproducibility:**

4: Could mostly reproduce the results, but there may be some variation because of sample variance or minor variations in their interpretation of the protocol or method.

**Reviewer Confidence:**

3: Pretty sure, but there's a chance I missed something. Although I have a good feel for this area in general, I did not carefully check the paper's details, e.g., the math, experimental design, or novelty.

---

> ### Author Rebuttal · Authors · 2023-08-27
>
> Thank you for your time and effort. We are grateful that you have identified that our work is interesting and the theoretical and experimental results are effective and promising. Given your strong rating and high confidence level, we respectfully ask you to engage in post-rebuttal discussions and help the rest of the reviewers converge to your opinion. We respond to your comments below.
>
>
> 1. We agree with your comments about LLMs. But as you mentioned the power of LLMs stems from pretraining on a large corpus. In other words, domain shift is less of a challenge when LLMs are used because it is more difficult to find a domain for which LLMs have not been exposed during pre-training. For this reason, this weakness is not specific to our method. Additionally, if we use LLMs in domains where that domain gap exists, we likely will observe performance degradation and UDA still will be relevant.
>
> 2. Thank you for pointing out the typos and errors. We will correct them and do a thorough pass to correct other errors and typos.
>
> We are thrilled that the reviewer has found our work to be strong. We respectfully ask the reviewer to engage in post-rebuttal discussions to help the reviewers with a less confident rating to reach an opinion with higher confidence.

---

### Official Review · Reviewer_Sndy · 2023-08-05

**Typos Grammar Style And Presentation Improvements:** 1.	Page 2, Line 132
**Soundness:** 3

**Excitement:**

3: Ambivalent: It has merits (e.g., it reports state-of-the-art results, the idea is nice), but there are key weaknesses (e.g., it describes incremental work), and it can significantly benefit from another round of revision. However, I won't object to accepting it if my co-reviewers champion it.

**Paper Topic And Main Contributions:**

The paper proposes a Gaussian mixture model (GMM) based method for domain adaptation in sentiment analysis. The idea is to maximize the distribution gap between classes in the GMM. This is achieved by sampling high confidence samples from the GMM to align the source and target domain distribution. Authors conducted experiments on Amazon product review dataset to evaluate the proposed method. Comparison with existing methods shows that the proposed method outperforms baselines when using tf-idf representation, and achieves comparable results when using pre-trained model embedding.

**Reasons To Accept:**

1.	The paper is addressing an important problem about domain adaptation in sentiment analysis. The general audience of the conference should be interested in this topic.
2.	The proposed method about maximizing the class distribution gap is theoretically sound.


**Reasons To Reject:**

1.	The proposed method aims at aligning parametric distributions in the embedding space. Authors demonstrate the effectiveness of the proposed method when using tf-idf and pre-trained model BERT. Empirical results show that BERT-based methods outperform tf-idf based approaches. However, when using BERT features, the proposed method can only achieve comparable results compared with BERT-DA. As a result, the proposed method is not convincing based on the experimental performance.
2.	The title of the paper is fairly mis-leading. The idea of the paper is to make use of the re-learning the GMM by sampling high confidence samples, so as to increase the gap between different class distributions. The proposed method does not deal with the representation of the data or feature embedding. The use of "internal representation" is confusing.
3.	Figure 3 is almost unreadable.


**Reproducibility:**

4: Could mostly reproduce the results, but there may be some variation because of sample variance or minor variations in their interpretation of the protocol or method.

**Reviewer Confidence:**

4: Quite sure. I tried to check the important points carefully. It's unlikely, though conceivable, that I missed something that should affect my ratings.

---

> ### Author Rebuttal · Authors · 2023-08-27
>
> Thank you for your time and effort. We are grateful that recognize the importance of the UDA problem in 2023 and also verify the theoretical soundness of our work. We hope that through continued discussion during the rebuttal period, we can address your concerns and convince you to reconsider your rating.
>
> It appears to us that by “not convincing based on the experimental performance”, you mean that we are not outperforming all baselines. We would like to bring your attention to a few points:
>
> Please note that in 5 tasks, we outperformed the baselines using BERT features and could get the best results.
>
> We know that a major reason for our lower performance in the remaining 7 tasks is that we could not achieve source-only BERT performances similar to Du et al. We are quite certain that if we could achieve similar source-free performance values, our UDA performance could be improved. This outcome is likely because of hyperparameter tuning but unfortunately, the code for Du et al is not public and the authors have not released their code. They also did not provide the hyperparameter values upon our contact. We tried to tune the hyperparameter but our resources are limited. Our code is available and we think regarding this disparity, our implementation is accessible and our reported numbers are much easier to confirm.
>
> In order to make a judgment about the competence of our method, comparisons should be based on all tasks and experiments. By considering experiments using TF-IDF features and 5 tasks with BERT features, I think we can conclude that our method leads to a convincing performance, and on average, our strengths outweigh the cases for which we are not getting SOTA performance.
>
> This concern can be easily addressed for the camera-ready version. We are more than happy to reconsider our title based on your feedback. Please recommend the title that might be a better choice and we are more than happy to incorporate your suggestions.
>
> We agree Fig 3 is very small which is due to the space limit. For review purposes, we made sure that it can be enlarged with good quality so it can be used by the reviewers to make a judgment. Fortunately, there will be a 9th page for the accepted papers and we can easily enlarge that figure to address your concern.
>
> In summary, we think that concerns 2 and 3 can be addressed straightforwardly. We hope you continue to engage in discussions about concern 1 and we hope upon addressing your concern you reconsider raising your rating because, with the current scores, it looks like your opinion can be very important for the final decision because you have the balancing scoring.

---

### Official Review · Reviewer_iCKc · 2023-08-06

**Typos Grammar Style And Presentation Improvements:** See questions.
**Soundness:** 3

**Excitement:**

2: Mediocre: This paper makes marginal contributions (vs non-contemporaneous work), so I would rather not see it in the conference.

**Missing References:**

Enrich references with those published after 2020.

**Paper Topic And Main Contributions:**

This paper presents a domain adaptation (DA) method, which is applied to cross-domain sentiment analysis. The proposed DA method makes contributions in two aspects, the first is aligning distributions of the source domain and target domain in an embedding space, and the second one is inducing larger margins between classes in the embedding space.  The first part is achieved by training an encoder to minimize domain discrepancy with the SWD (Lee et al., 2019) method. The second one is based on learning an intermediate parametric distribution by estimating it as a Gaussian mixture model (GMM).

**Questions For The Authors:**

(1) There are no references after 2021, raising concerns about the correctness of some claims in this paper. For example, the discrepancy between different domains stems from different usage of terms and phrases (L33), does this domain gap still a major issue in neural models, given NN models can learn context instead of focusing too much on terms and phrases.
Another example is BERT features has not been used extensively on domain adaptation tasks (L467), is this claims still true in 2023, as there are no recent references since 2021.
(2) The figures in the paper are too small, they should be clear by using more space. The paper can make more space from Section 6.1 with a more concise presentation.
(3) The introduction and related work are mixed together and difficult to understand the contribution of this paper.
(4) Some citations such as L98-102 are difficult to understand. Using WD for domain alignment boosts performance in visual domains is introduced by Damodaran et al. 2018 or other citations in the same sentence?
(5) There are some grammar errors such as "a large enough labeled data points" in L163.
(6) This paper reimplements vanilla BERT and reports a different result with the same model in (Du et al., 2020) with a large margin, which raises concerns about the reimplementations.

**Reasons To Accept:**

This paper presents a novel domain adaptation method and uses experiments to assess its effectiveness. It seems to be effective in tfidf setting, and self-implemented BERT.

**Reasons To Reject:**

The writing of this paper makes it difficult to understand.

**Reproducibility:**

2: Would be hard pressed to reproduce the results. The contribution depends on data that are simply not available outside the author's institution or consortium; not enough details are provided.

**Reviewer Confidence:**

2: Willing to defend my evaluation, but it is fairly likely that I missed some details, didn't understand some central points, or can't be sure about the novelty of the work.

---

> ### Author Rebuttal · Authors · 2023-08-27
>
> Thank you for your time and effort. We are glad that you have recognized the novelty of our work. We are also grateful that you honestly declared your confidence level as in our experience, many reviewers tend to overestimate their confidence. Given your confidence level and also the opinion of the rest of the reviewers, we would like to respectfully ask you to give us a second chance by elaborating on our response and engaging in post-rebuttal discussions. We also think that if you check the references on UDA for sentiment analysis and compare our work side-by-side, you can evaluate our work more meaningfully. We hope you find our responses below convincing:
>
> Thank you for bringing to our attention that we have used works published prior to 2021. This has not been intentional. Upon your point, we did a more expansive search and found the following papers for comparison::\
>
> Long, Q., Luo, T., Wang, W. and Pan, S., 2022, July. Domain Confused Contrastive Learning for Unsupervised Domain Adaptation. In Proceedings of the 2022 Conference of the North American Chapter of the Association for Computational Linguistics: Human Language Technologies (pp. 2982-2995). (Table 1 and Table 2)
>
> Badr, H., Wanas, N. and Fayek, M., 2022. Unsupervised domain adaptation with post-adaptation labeled domain performance preservation. Machine Learning with Applications, 10, p.100439. (Table 3)
>
> Ryu, M., Lee, G. and Lee, K., 2022. Knowledge distillation for bert unsupervised domain adaptation. Knowledge and Information Systems, 64(11), pp.3113-3128. (Table 1)
>
> Huang, L., Zhao, W., Liu, Y., Yang, D., Liew, A.W.C. and You, Y., 2023. An Evidential Multi-Target Domain Adaptation Method Based on Weighted Fusion for Cross-Domain Pattern Classification. IEEE Transactions on Neural Networks and Learning Systems. (Table VI)
>
>
> There are also several post-2021 published works on “source-free UDA” and “multi-source UDA” for sentiment analysis.  There are also several  2023 unpublished manuscripts on Arxiv. You can easily find these works on Google Scholar. We think that from these works, we can conclude:
>
> UDA has not been resolved in 2023 and still ongoing research continues to address the challenge of the “domain gap” for sentiment analysis. LLMs have made progress to mitigate this challenge but it is not the case that they have solved the challenge completely.
>
> If you compare our results with the tables, referenced in each work in the above in parentheses, our work remains compelling which demonstrates that our results are still competitive.
>
> We hope you agree that the ongoing research after 2021 demonstrates that the challenge of UDA for sentiment analysis is still an active research area, despite advances in NN. We will also include the above works in our discussion. If you think there are other missing references, we are more than glad to include them if you give us pointers.
>
> 2.  This concern can be easily addressed in the camera version because there will be a 9-page limit and we can easily make the figures larger and nicer. So, we can do this task even without making section 6.1 more concise.
>
> 3. If you provide more guidance, we are more than happy to improve the Introduction and the Related Work section given the extra space that we will have. We kindly ask you to provide more details so we know how we can make the paper more clear in these sections and reduce unclarities.
>
> 4. We agree we could be more clear and currently that sentence is ambiguous. All those references have explored using WD for domain adaptation in visual domains. We will make that sentence more clear
>
> 5. Thank you for pointing out. We will correct it and do a thorough pass to correct grammatical errors and typos. If you find other errors and typos, please let us know.
>
> 6. You are correct but we would like to say please first note that our results are close. It is not the case that we are getting totally irrelevant results. Second, the results can depend on hyperparameters and we did our best to find the best hyperparameters to report the best results but our GPU resources are limited and we could not perform an extensive search. We also contacted the authors but unfortunately, we were not given the hyperparameter values used by the authors. In conclusion, we think the disparity is within an acceptable statistical margin. More importantly, please note that as opposed to Du et al, our code is available and you can easily check our implementation. We think that if there are concerns about disparity, our results are much easier for verification because our code is available. In contrast, when the code is not released and the hyperparameters are not provided in a paper, it is not easy to regenerate the results.
>
> We respectfully ask the reviewer again, to read our responses, check the literature, read the rest of the reviews, engage in post-rebuttal discussion, and if convinced reconsider the final rating. We hope through continued discussions, we can address all the raised concerns.

---

### Meta-Review · Area_Chair_Uwiw · 2023-09-17

**Recommendation:** 3

**Metareview:**

The paper proposes a great theoretical solution and theories and proofs are well explained, however, in practice the results are comparable or superior only with traditional encoding methods like tf-idf, other results where more advanced encoding methods like BERT are used other approaches mostly outperform the proposed methods.

Although the proposed method is used in 4 different cross-domains it is still done on a small set of benchmarked datasets, hence, it makes the effectiveness of the method questionable. Sentiment is very subjective and such a theoretical method should be tested on a big dataset instead.

---

### Decision · Program_Chairs · 2023-10-07

**Decision:**

Accept-Findings

**Comment:**

The paper proposes a great theoretical solution and theories and proofs are well explained, however, in practice the results are comparable or superior only with traditional encoding methods like tf-idf, other results where more advanced encoding methods like BERT are used other approaches mostly outperform the proposed methods.

Although the proposed method is used in 4 different cross-domains it is still done on a small set of benchmarked datasets, hence, it makes the effectiveness of the method questionable. Sentiment is very subjective and such a theoretical method should be tested on a big dataset instead.